# Neurocognitive Outcomes After Extracranial Surgery and General Anesthesia in Patients with a History of Mild-to-Moderate Traumatic Brain Injury: Systemic Review and Meta-Analysis

**DOI:** 10.3390/biology14060640

**Published:** 2025-05-31

**Authors:** Zeeshan A. Khan, Tahiris A. Duran, Dewan Md. Sumsuzzman, Ling-Sha Ju, Christoph N. Seubert, Anatoly E. Martynyuk

**Affiliations:** 1Department of Anesthesiology, College of Medicine, University of Florida, JHMHC, 1600 SW Archer Road, P.O. Box 100254, Gainesville, FL 32610-0254, USA; zkhan2@anest.ufl.edu (Z.A.K.); durantahiris@ufl.edu (T.A.D.); lju@anest.ufl.edu (L.-S.J.); cseubert@anest.ufl.edu (C.N.S.); 2Agent-Based Modelling Laboratory, York University, Toronto, ON M3J 1P3, Canada; dewanp@yorku.ca; 3McKnight Brain Institute, College of Medicine, University of Florida, Gainesville, FL 32610-0254, USA

**Keywords:** postoperative neurocognitive disorder, traumatic brain injury, general anesthesia

## Abstract

This study investigated whether individuals with mild-to-moderate traumatic brain injury (TBI) are at increased risk of neurocognitive deficits following surgeries that do not involve the brain under general anesthesia (GA). We have systematically reviewed and meta-analyzed observational studies, finding that TBI patients who underwent these surgeries performed worse on cognitive tests compared to those who did not undergo GA surgeries. However, the length of hospital stays, and overall recovery were not affected. Since only a few studies were deemed eligible for this meta-analysis, more research is needed to understand the full impact of these surgeries on TBI patients.

## 1. Introduction

General anesthesia and surgery (GA/surgery), while essential for many medical procedures, can be associated with various adverse effects. Among these, accelerated neurocognitive decline following GA/surgery, also known as postoperative neurocognitive dysfunction, represents a significant public health concern [1]. Postoperative neurocognitive dysfunction can present as an acute cognitive impairment called postoperative delirium, typically arising within the first few days after GA/surgery [1]. The hallmark symptoms of postoperative delirium include confusion, agitation, delusions, and deficits in orientation and attention [1,2]. This condition is linked to prolonged hospital stays and increased mortality. Some patients may also experience delayed neurocognitive recovery or temporary cognitive impairments, such as difficulties with memory, attention, and executive function, which can last up to one month after GA/surgery. Beyond these short-term impairments, neuropsychological tests may reveal persistent declines in cognitive domains, including memory, awareness, reasoning, judgment, and language in certain patients during the post-GA/surgery period, extending for a year or longer. These long-term impairments are classified as postoperative neurocognitive disorder (PND) [3,4]. Patients with PND may face challenges in daily activities, ultimately reducing their quality of life. Moreover, evidence suggests that PND may increase the risk of developing dementia [5,6].

The mechanisms underlying PND and the populations most vulnerable to it remain poorly understood. Current data indicate that advanced age and preexisting neurodegenerative conditions, which are associated with upregulated inflammation and stress, are key factors contributing to vulnerability to PND. Since neurodegenerative conditions, inflammation, and stress tend to worsen with age, it is unsurprising that PND is most commonly diagnosed in the aging population [7]. However, neurodegeneration, inflammation, and stress are not exclusive to the elderly. Younger individuals may also experience these conditions across a broad spectrum of pathophysiological states. One such state is traumatic brain injury (TBI). Although TBI affects individuals of all ages, younger adults engaged in high-risk activities, such as contact sports or military service, are among those most frequently affected [8]. TBI can trigger progressive brain cell damage, potentially leading to conditions like chronic traumatic encephalopathy [9]. Following TBI, the brain often experiences prolonged inflammation, exacerbating tissue damage and contributing to cognitive decline [10]. TBI can also lead to chronic stress responses, both physiological and psychological, including elevated cortisol levels and a heightened risk of mental health conditions such as anxiety and depression [11]. Altogether, these TBI-initiated responses can impair memory, attention, executive function, and problem-solving abilities, potentially increasing the risk of neurodegenerative diseases such as Alzheimer’s or Parkinson’s [12,13,14,15].

TBI patients often require surgeries under GA that are unrelated to their brain injury. For instance, they may need procedures to address peripheral injuries sustained during the same traumatic event or unrelated to TBI, such as fractures or internal organ damage [16,17,18]. Notably, studies suggest that undergoing non-TBI-related surgeries after TBI may potentially affect recovery from TBI-induced abnormalities, including impairments in cognitive and functional abilities [19]. Collectively, this evidence suggests that patients with a history of TBI may be more susceptible to GA/surgery-induced PND due to the interaction of TBI- and GA/surgery-associated inflammation, stress, neurodegeneration, and weakened cognitive reserves [20,21,22]. To explore this issue, we systematically reviewed and meta-analyzed existing clinical studies to assess whether patients with a history of TBI undergoing non-brain-related peripheral surgeries under GA (extracranial surgeries) are at a higher risk of developing post-GA/surgery PND.

## 2. Materials and Methods

The protocol of this systematic review and meta-analysis was registered at PROSPERO (CRD42024510980) and published prior to the review’s completion [23].

### 2.1. Information Sources and Search Strategy

The following databases were searched to retrieve articles from inception to April 2024 (Appendix A) without language restrictions: Ovid Medline, Ovid Em-base, Ovid Emcare, Global Health, and APA PsycINFO. Our search strategy utilized Boolean operators (AND/OR) and relevant keywords, such as “TBI”, “anesthesia”, and “extracranial surgery”. Additionally, the reference lists of eligible studies meeting our inclusion criteria were manually searched to identify additional relevant studies. The full search strategy for all databases is provided in Appendix A and published in our protocol [23].

### 2.2. Inclusion and Exclusion Criteria

#### 2.2.1. Study Design

Observational studies were used in this meta-analysis to examine PND in patients with TBI who underwent extracranial surgeries under GA. Reviews were excluded, as they do not provide new primary data.

#### 2.2.2. Population/Participants

Our study included patients with mild-to-moderate TBI who underwent extracranial surgery under GA. TBI severity was determined using the Glasgow Coma Scale (GCS), with scores of 13–15 indicating mild TBI and 9–12 indicating moderate TBI [24]. Patients with severe TBI (GCS < 8) were excluded, as they often require intracranial surgery for treatment [25]. Non-clinical and animal studies were also excluded. 

#### 2.2.3. Intervention/Exposure

We included studies in which patients had mild-to-moderate TBI and underwent extracranial surgery under GA. Cardiac and pulmonary bypass surgeries were also excluded, as they may further exacerbate TBI-like symptoms [26,27].

#### 2.2.4. Control

The control group consisted of patients with mild-to-moderate TBI who had not undergone any form of surgery. Patients with severe TBI were excluded from the control group.

#### 2.2.5. Outcomes

The primary outcomes evaluated were long-term neurocognitive functions using trail-making tests A and B (TMT-A/B). For post-operative quality-of-life evaluation, Glasgow Outcome Scale—Extended (GOSE), and length of ICU and hospital stay (LOS) were analyzed as outcomes, as they are frequently used indicators of functional recovery following surgery [28,29,30].

### 2.3. Study Selection

At least two independent reviewers (Z.A.K. and T.A.D.) screened and selected studies, agreeing on which articles to include in the final analysis. In cases of disagreement, additional reviewers (L.S.J. and D.M.S.) were consulted to determine whether a study should be included or excluded. Data were collected manually after screening, with duplicates initially removed using the Rayyan software (Cambridge, MA, USA). Two researchers reviewed and agreed upon relevant articles, while the remaining irrelevant dataset was processed through the ASReview software (Version 1.6.2, Utrecht, The Netherlands) to train the machine-learning model and further filter relevant studies [31,32].

Since the ASReview software can only be used by one person at a time, the two independent researchers alternated during the screening process. Researcher 1 (Z.A.K.) screened 400 articles before transferring the task to Researcher 2 (T.A.D.), who completed the process. In total, the researchers screened 20% of the data (n = 19,129), starting with the most relevant and proceeding to the least relevant, as suggested by ASReview. On the other hand, a previous researcher had successfully identified all of the relevant literature by screening only 15% of articles [31]. Since 20% of 19,129 is 3825.8, we have screened 3826 articles. Once over 500 articles had been reviewed since the last relevant paper was found, screening was concluded (Figure 1). Relevant outcome measures were compiled from each article and used for data analysis. The selected studies are stored in a University of Florida shared drive for article management.

### 2.4. Data Extraction

We collected data on first authors, year of publication, patient age, gender, type of anesthesia, country of study origin, timing of cognitive assessment after surgery and anesthesia, comorbidities, diagnosis, and exposure. Plot Digitizer, an open-source software, was used to extract data presented in graphical form within the articles [33].

### 2.5. Risk-of-Bias Assessment

The Newcastle–Ottawa scale (NOS) was used to evaluate the risk of bias in the observational cohort studies included in our meta-analysis [34,35]. The NOS assesses studies across three domains: selection, comparability, and outcome assessment. Each domain contains a set of criteria, and studies are awarded stars based on their adherence to these criteria. Studies were classified as poor (0–3 stars), fair quality (4–6 stars), or good quality (7–9 stars) based on the number of stars awarded [34].

### 2.6. Data Synthesis and Statistical Analysis

In our meta-analysis, we thoroughly analyzed various types of data, including both continuous and binary outcomes, using appropriate statistical methods. For continuous outcomes, we applied the mean difference (MD) using a random-effects model when studies measured outcomes on the same scale, ensuring direct comparison [36]. When studies used different scales, we utilized the standardized mean difference (SMD) to standardize the data [36]. Effect sizes were evaluated according to Cohen’s guidelines: small (0.2), medium (0.5), and large (0.8) [37]. For binary outcomes, we calculated the odds ratio (OR), with pooled estimates back-transformed for better interpretability and presented in forest plots. For single-group binary data, we converted raw event counts into proportions before pooling to standardize the data [38].

The I^2^ statistic was used to assess heterogeneity, ensuring that conclusions were based on a comprehensive synthesis of the data. Heterogeneity was interpreted as follows: 0–25% indicates low heterogeneity, 26–50% suggests moderate heterogeneity, and above 50% reflects high heterogeneity [39]. To ensure the robustness of our results, sensitivity analyses included excluding individual studies one at a time, removing high-risk studies, and evaluating the impact of single study. To minimize publication bias, we utilized funnel plots and the Egger regression test to detect and address potential biases [40]. All statistical analyses were conducted using the metaphor package in R (version 1.6.2) or Stata/SE (version 16), with a significance threshold set at *p* < 0.05. Continuous outcomes were reported using MD/SMD with 95% confidence intervals (CIs).

## 3. Results

### 3.1. Study Search and Selection

A total of 27,139 articles were retrieved from our selected databases using our curated search strategy (Appendix A). Utilizing the Rayyan screening tool, 8010 studies were identified as duplicates and removed from the initial dataset. The remaining 19,129 studies were screened using the artificial intelligence tool ASReview [31,32]. Based on ASReview’s ranking of relevant articles, the reviewers manually screened the top 20% of the most relevant studies by title and abstract, totaling 3826 articles. Notably, more than 500 of the lowest-ranked articles within this set were found to be irrelevant according to the selection criteria (Figure 1). Of the 3826 screened articles, 29 were selected for full-text review, while the remaining 15,303 (19,129 − 3826 = 15,303) were deemed ineligible without title and abstract screening. The full texts of these 29 selected articles were assessed against the selection criteria, resulting in the exclusion of 12 studies due to unrelated populations, 2 due to irrelevant study design, 6 due to irrelevant outcomes, and 4 due to the absence of an appropriate control group (Appendix A). Ultimately, five studies were included in the review [41,42,43,44,45] (Figure 2). Earlier studies have reported that screening only 15% of the most relevant articles is sufficient to successfully identify all relevant literature [31].

### 3.2. Study Characteristics

The main characteristics of the included studies are summarized in Table 1. Notably, only five relevant studies were identified over the last 27 years between 1997 and 2024 [41,42,43,44,45]. One study was excluded as it has mixed data of moderate and severe TBI [46]. Among the selected studies, two involved lower-limb surgery [41,43], one focused on rib fracture [44], one examined laparoscopic abdominal surgery [42], and one encompassed heterogeneous extracranial surgical interventions [45]. All studies assessed TBI severity using the GCS, and the Injury Severity Score (ISS)/Abbreviated Injury Scale (AIS) to evaluate the severity of multisystem injuries. Both male and female subjects were included in all studies; however, the majority of participants in both the experimental and control groups were male, except in [43], where gender distribution was equal. The mean age of study participants was similar across all studies, with reported values of 37.5, 35.6, 42.7, 50, and 42.2 in [41], [42], [43], [44], and [45], respectively. Three of the selected studies originated from the United States [42,43,45], one from Canada [41], and one collected data from hospitals internationally [44]. Neurocognitive outcome measures varied across the included literature, though some overlapping measures were incorporated in this study. TMT-A/B and GOSE were used in [41,45]. Quality of life outcomes, such as hospital LOS, were reported by [41,42,43,44]. Additionally, intensive care unit (ICU) LOS was reported by [42,44,45]. The follow-up periods for neurocognitive testing varied among the studies: one study [45] included assessments at both 2 weeks and 6 months, [41] had an average follow-up of 32 months, and [43] reported a follow-up duration of 12 months. The last follow-up period from each study was selected as the data point for meta-analysis. None of the included studies contain baseline neurocognitive data for TBI patients prior to anesthesia or surgery. Regarding confounding variables or preexisting conditions, the included studies excluded patients with prior neurocognitive deficiencies or a history of TBI. 

### 3.3. Risk of Bias and Quality Reporting

Figure 3 presents the results of the Newcastle–Ottawa quality assessment scale for each selected study. All studies demonstrated a low risk of bias, with [41] receiving the lowest score of 7 out of 9 stars. Therefore, all studies were classified as good or high quality for this analysis, indicating a low risk of bias.

### 3.4. Meta-Analysis

#### 3.4.1. Trail-Making Tests A and B

Two studies reported using the TMT-A test to assess rote memory in individuals with TBI exposed to GA compared to unexposed individuals [41,45]. In one study [41], the mean age of the subjects was 37.5 years at the time of evaluation, with a mean interval of 32 months between exposure and evaluation (minimum follow-up of 12 months). In the other study [45], the mean age was 42.2 years, with a 6-month interval between exposure and evaluation. In our meta-analysis of both studies, a significant difference was observed in the time required to complete TMT-A between exposed and unexposed TBI patients [I^2^ = 0.0%, MD = 2.04; 95% CI (0.38, 3.70); *p* = 0.016] (Figure 4A). It is important to note that rote memory refers to the ability to recall information through repetition without necessarily understanding its meaning or context, whereas the primary purpose of the TMT-A is to evaluate processing speed, attention, and visual–motor coordination [47,48,49]. However, some investigators suggest that the processing speed, visual attention, sequencing, and motor function assessed by the TMT-A can be interpreted as features of rote memory, given that the test requires, for example, the repetitive and sequential connection of numbers under timed conditions [49,50].

The aforementioned studies also employed the TMT-B test to compare executive function between the two groups [41,45]. A statistically significant difference in completion time, typically measured as the mean difference (MD), or the average difference in completion time between two groups, was also found between them [I^2^ = 0.0%, MD = 16.59; 95% CI (9.58, 23.60); *p* < 0.001] (Figure 4B). While there is no universally agreed-upon MD threshold for clinical significance, an MD of several seconds (e.g., 10–20 s or more) may indicate meaningful cognitive impairment in TBI populations [51].

#### 3.4.2. Glasgow Outcome Scale

Two studies [41,45] reported on using the GOSE to assess patient outcomes. One study [45] utilized a complete set of the GOSE evaluation, whereas the other [41] reported only proportions of good–moderate scores. Therefore, the OR was used for the meta-analysis. In [45], GOSE scores between 5 and 8 were classified as good–moderate outcomes. We found an OR of 0.475, which suggests that patients in the TBI group exposed to GA/surgery had a lower odds of achieving a good–moderate GOS score, though the results were not statistically significant (Figure 4C). Additionally, high heterogeneity was observed [I^2^ = 93.3%].

#### 3.4.3. Length of ICU and Hospital Stays

Three articles focused on the duration of intensive care: [42,44,45]. We found no significant difference in ICU LOS [I^2^ = 81.4%, MD = 1.94; 95% CI (−1.25, 5.13); *p* = 0.137] days (Figure 5A). Regarding hospital LOS, the data from [41,42,43,44] were used for our analysis. Using a random-effects model, we found no significant difference in hospital LOS [I^2^ = 23.1%, MD = 0.99; 95% CI (−1.59, 3.57); *p* = 0.45] days (Figure 5B).

A funnel plot and Egger regression test suggested a low risk of publication bias for hospital LOS [*p*-value = 0.7680] (Figure 6).

#### 3.4.4. Sensitivity Analysis

Sensitivity analysis excluding each individual study at a time showed a significant effect only when Prins et al. [44] was removed from the hospital LOS and ICU LOS analyses (Appendix A). For ICU LOS, excluding [44] resulted in a significant *p*-value; however, heterogeneity remained high, with only a slight reduction [MD = 3.43; CI [1.00, 5.85]; *p* = 0.01; I^2^ = 80.30%] compared to the original results [MD = 1.94; CI [−1.25, 5.13]; *p* = 0.137; I^2^ = 81.40%]. No single study exclusion reduced the heterogeneity below 50%. Regarding hospital LOS, the *p*-value remained insignificant, but heterogeneity was eliminated [MD = 2.14; CIs [−0.38, 4.65], *p* < 0.001; I^2^ = 0.00%] compared to the original results [MD = 0.99; CIs [−1.59, 3.57], *p* = 0.234; I^2^ = 38.60%]. Excluding other individual studies did not yield significant effects on hospital LOS or ICU LOS.

## 4. Discussion

This study systematically reviewed and meta-analyzed published observational research to assess whether individuals with a history of mild-to-moderate TBI are at an increased risk of developing PND following extracranial surgeries under GA. Only five relevant articles were identified among studies involving TBI patients published between 1997 and 2024. The meta-analysis findings, based on this limited number of studies, suggest that patients with a history of mild-to-moderate TBI may have a higher likelihood of developing neurocognitive complications after undergoing extracranial surgery under GA compared to patients with similar TBI histories who have never undergone surgery under GA. Specifically, this meta-analysis found that patients with a history of mild-to-moderate TBI who underwent extracranial surgeries under GA demonstrated a worse performance on the TMT-A and TMT-B tests. Notably, the findings of this meta-analysis suggest that surgeries under GA in patients with mild-to-moderate TBI impacted neurocognitive function but not general functional recovery. Thus, these patients did not differ from those with TBI of similar severity who had not undergone surgery or GA in terms of ICU LOS and overall hospital LOS. Furthermore, although the GOSE data showed high heterogeneity and no significant differences between the study and control groups, it demonstrated a trend of improvement in patients with mild-to-moderate TBI who underwent extracranial surgery under GA compared to the control group. 

The delayed completion time observed in both TMT-A and TMT-B for patients with TBI exposed to extracranial surgery under GA suggest potential impacts on cognitive flexibility, visual attention, and task-switching abilities [52]. TMT-A primarily assesses fundamental cognitive processes, including processing speed, attention, and visual search capabilities, requiring participants to connect numbered circles in ascending order as quickly as possible [48,52]. The delayed completion time in TMT-A indicates potential impairments in these areas, which are crucial for general cognitive performance and daily functioning [48,52,53]. Performance in TMT-A forms the foundation for the more complex assessments conducted in TMT-B. 

TMT-B, on the other hand, evaluates higher-order cognitive functions such as mental flexibility, task-switching, and executive functioning [54]. Participants alternate between numbers and letters in ascending order, introducing a switching component [55]. This test is particularly valuable for assessing abilities like planning, problem solving, and multitasking, which are often compromised in individuals with brain injuries or neurodegenerative conditions [48,54,55,56]. Moreover, TMT-B is more sensitive than TMT-A in detecting subtle cognitive deficits, especially those linked to frontal lobe functions [57]. The results of our meta-analysis align with the recent evidence from the TRACK-TBI study, which reported that exposure to surgery under GA is associated with worse neurocognitive outcomes, particularly in executive functioning, as measured by TMTs [45]. The neurocognitive outcomes observed in patients with TBI who underwent extracranial surgery under GA indicate significant impairments in various cognitive domains. Specifically, the delayed completion times in both TMT-A and TMT-B suggest deficits in cognitive flexibility, visual attention, and task-switching abilities [45]. Additionally, impairments in memory retention, language processing, and verbal fluency suggest that the effects of surgery and anesthesia may extend beyond executive functioning and cognitive flexibility.

Our meta-analysis did not reveal a significant difference between the control and treatment groups in terms of ICU LOS and hospital LOS. ICU LOS reflects the intensity and duration of care required in a critical setting, highlighting complications such as intracranial pressure management or ventilatory support [58]. Hospital LOS, on the other hand, represents the broader recovery trajectory, encompassing stabilization, medical treatments, and rehabilitation needs [59]. Therefore, our findings suggest that, in contrast to neurocognitive functioning, surgeries under GA in patients with mild-to-moderate TBI may have a minimal-to-no effects on general functioning recovery. The notion that surgeries under GA predominantly exacerbate neurocognitive impairment, a form of PND, rather than affecting general functional recovery are also supported by the results of this meta-analysis of the GOSE data. Our analysis of the GOSE data found no significant difference between the treatment and control groups, and even a trend towards improvement in TBI patients with surgeries under GA. The GOSE is more focused on evaluating functional recovery and general outcomes following TBI, such as independence, social participation, and daily living abilities [60,61]. While it provides valuable insights into a patient’s overall recovery and quality of life, it is not specifically designed to assess neurocognitive function [60,61]. 

Over the last 27 years, only five relevant studies were identified that investigate the effect of extracranial surgery under GA on neurocognitive outcomes (PND) in patients with mild-to-moderate TBI, highlighting the scarcity of comprehensive research on this topic. This is particularly concerning given that millions of individuals experience TBI annually [62,63]. Mild-to-moderate TBI is especially common among younger individuals, particularly those involved in contact sports or military service [64]. The limited research on PND in TBI patients may partly stem from the fact that TBI frequently affects younger individuals, whereas the phenomenon of PND has historically been associated with older age [65,66]. However, the results of our meta-analysis, particularly the findings from [41] and [45], whose studies involved participants aged 37.5 and 42.2 years, respectively, demonstrate through TMT-A and TMT-B test results that PND in TBI patients is not restricted to older individuals. This underscores the need for more comprehensive research in this field. Furthermore, another recent study reports the intricate interplay between surgery, anesthesia, and TBI outcomes, emphasizing the need for a careful risk–benefit analysis when considering surgical interventions for TBI patients [67].

We hope that this systematic review and meta-analysis will galvanize much-needed clinical and laboratory research on this clinically and scientifically significant topic, which has so far lacked sufficient attention, as evidenced by the identification of only five broadly relevant studies published in this field over the past 27 years.

Apart from identifying a significant lack of literature on PND in patients with TBI, the available studies revealed high unknown heterogeneity in the GOSE and ICU LOS. This heterogeneity in clinical studies can be attributed to multifactorial causes, including variations in surgical interventions, anesthesia regimens, patient demographics, and environmental factors. While the confounding effects of many of these factors can be mitigated through standardized study designs, addressing the influence of demographic and environmental variables in large patient population studies, where preexisting conditions are often patient-specific, remains challenging.

Such challenges may be addressed, particularly in the initial stages of investigation, through preclinical animal models that allow for rigorous control of confounding factors. Using these models, we demonstrated that not only can young adult rats with TBI be vulnerable to PND, but their future offspring, who are TBI- and surgery-GA-naïve, may also develop neurobehavioral abnormalities. The latter finding is especially alarming as epidemiological studies find that the children of parents with TBI are more likely to develop psychiatric disorders [68,69,70,71,72,73].

## 5. Conclusions

In summary, the findings of this systematic review and meta-analysis of published observational research suggest that patients with mild-to-moderate TBI are at an increased risk of developing PND following extracranial surgery under GA. Additionally, the results indicate that while extracranial surgery under GA may exacerbate neurocognitive functioning in TBI patients, it has minimal impact on their overall functional recovery. This review and meta-analysis also underscores the limited number of studies on PND in TBI patients, highlighting an urgent need for further clinical and preclinical research on this critical topic.

## Figures and Tables

**Figure 1 biology-14-00640-f001:**
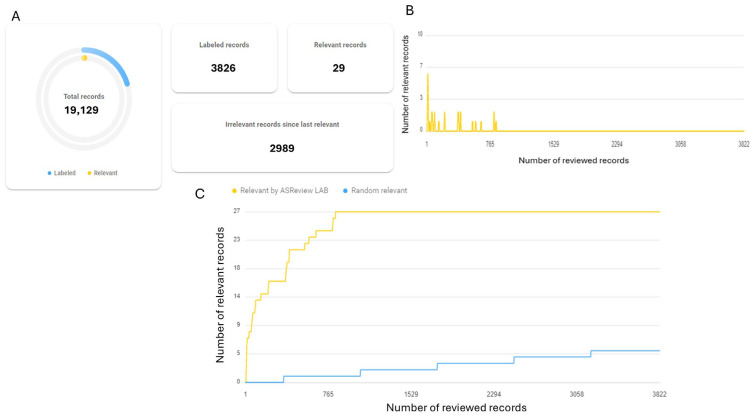
Screening of identified articles was conducted using the artificial intelligence (AI)-driven ASReview tool. (**A**) ASReview chart quantifying the number of reviewed records compared to the total records. (**B**) ASReview graph illustrates how many records were marked as relevant over the course of the review. Screening was discontinued after hundreds of records were searched without identifying additional relevant articles. (**C**) ASReview demonstrating how AI enhances the efficiency of finding relevant articles compared to the random ordering of searchable records.

**Figure 2 biology-14-00640-f002:**
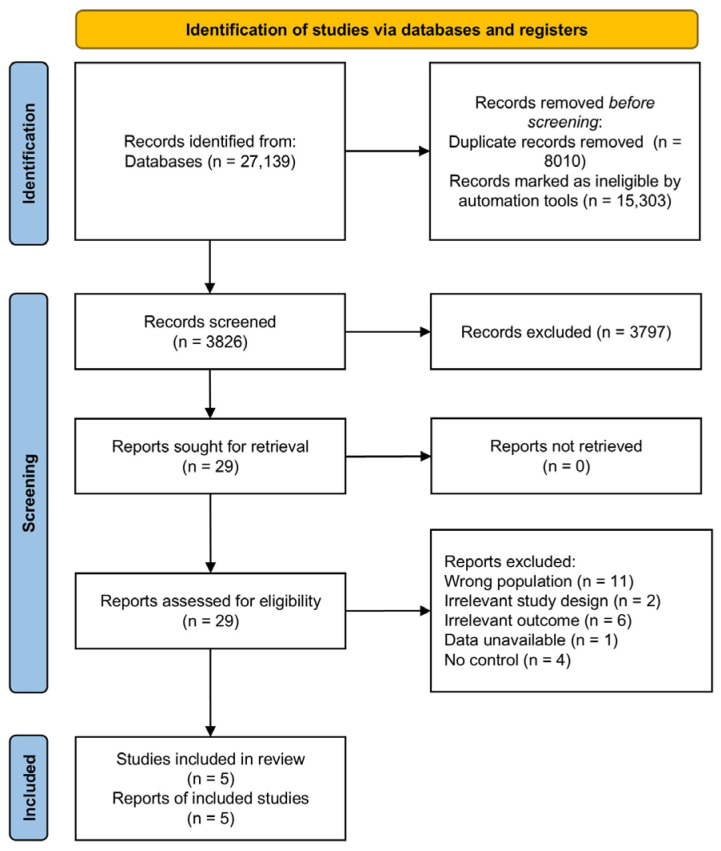
The PRISMA 2020 flow diagram illustrates the process of identifying, screening, and including studies in a meta-analysis. A total of 27,139 articles were retrieved from our selected databases using our curated search strategy (Appendix A). Utilizing the Rayyan screening tool, 8010 studies were identified as duplicates and removed from the initial set of 27,139 articles. The remaining 19,129 studies were screened using the artificial intelligence tool ASReview [31,32]. Based on ASReview’s ranking of relevant articles, the reviewers manually screened the top 20% of the most relevant studies by title and abstract, totaling 3826 articles. Of these, 29 were selected for full-text review, while the remaining 15,303 articles (19,129 − 3826 = 15,303) were deemed ineligible without title and abstract screening. The full texts of the 29 selected articles were assessed against the selection criteria, leading to the exclusion of 12 studies due to unrelated populations, 2 due to irrelevant study design, 6 due to irrelevant outcomes, and 4 due to the absence of an appropriate control group. Ultimately, five studies were included in the review [41,42,43,44,45].

**Figure 3 biology-14-00640-f003:**
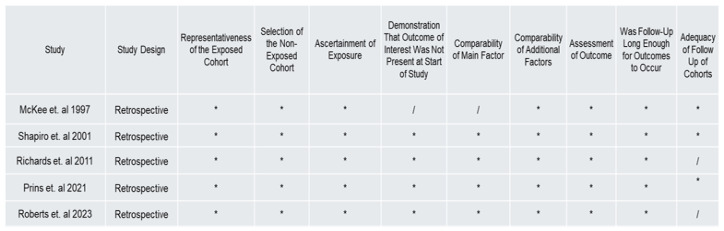
Newcastle–Ottawa coding quality assessment results were used to evaluate the risk of bias for each included study [41,42,43,44,45] (reference number are from top to bottom of column). Under each criterion, an article was awarded “*” if it met the criteria, or “/” if it felt short in that category. Scoring was based on the number of “*”s out of a possible 9. A higher number of “*”s indicated a lower risk of bias.

**Figure 4 biology-14-00640-f004:**
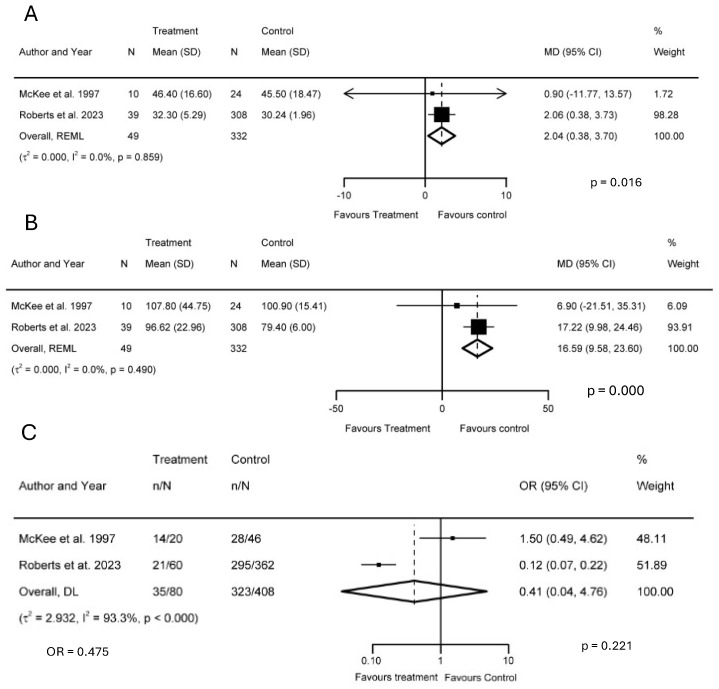
Forest plot comparing differences in (**A**) Trail-making test A (TMT-A), (**B**) Trail-making test B (TMT-B), and (**C**) Glasgow Outcome Scale (GOS) scores using odds ratios [41,45]. The prism represents the overall statistical results of the experimental data, squares indicate the weight of each study, and horizontal lines represent the 95% CIs for each study. The *p*-values next to I^2^ on the right side of the forest plot represent the significance of the heterogeneity, while the *p*-values on the left side show the significance between the treatment and control groups. Abbreviations: TBI, traumatic brain injury; I^2^, heterogeneity; MD, mean difference; CI, confidence interval; SD, standard deviation; OR, odds ratio.

**Figure 5 biology-14-00640-f005:**
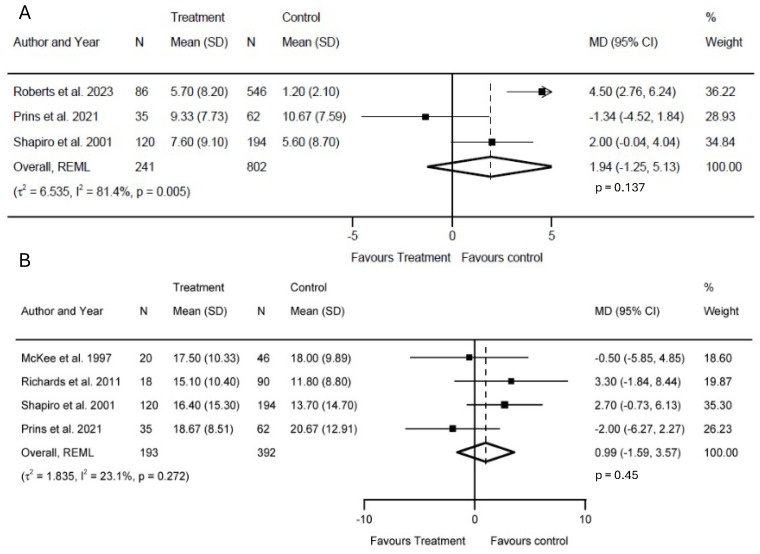
Forest plot comparing length of stay in the (**A**) ICU [42,44,45] and (**B**) hospital [41,42,43,44] between TBI patients who underwent extracranial surgery/GA and those who did not undergo surgery/GA. The prism represents the overall statistical results of the experimental data, squares indicate the weight of each study, and horizontal lines represent the 95% CIs for each study. The *p*-values next to I^2^ on the right side of the forest plot represent the significance of the heterogeneity, while the *p*-values on the left side show the significance between the treatment and control groups. Abbreviations: TBI, traumatic brain injury; ICU, intensive care unit; HOS, hospital; I^2^, heterogeneity; MD, mean difference; CI, confidence interval; SD, standard deviation. The arrow denotes a capped upper bound of the 95% CI.

**Figure 6 biology-14-00640-f006:**
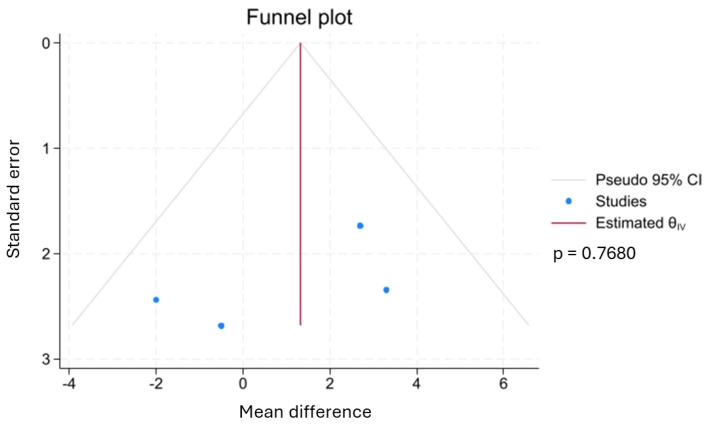
Funnel plot suggesting a low risk of publication bias in length of hospital stay analysis.

**Table 1 biology-14-00640-t001:** Study characteristics (n/a = not available).

Author, Year and Country	Age (Years)	BMI/Body Weight (Kg)	Gender	Type of Surgery	Route of Administration	Anesthetics Used	Outcome	Duration After Which Neurocognitive Tests Were Assessed	**Technique/Scale**	**Journal**
McKee et al., 1997, Canada [41]	16–63 (Mean = 37.5)	n/a	Male (Study Group): 35 Female (Study Group): 11 Male (Control): 75 Female (Control): 24	EC (orthopedic) -femoral fracture	n/a	n/a	Executive functioning, reasoning, problem-solving, visual motor attention, cognitive impairment	avg 32 months (range 12–59 months)	Return to work status, Glasgow Outcome Scale (GOS), the Category test, and the Trails A and B tests	The Journal of Trauma: Injury, Infection, and Critical Care
Shapiro et al., 2001, USA [42]	35.6	n/a	n/a	laparaotomy (intra-abdominal)	n/a	n/a	Quality of life (postoperative recovery), Injury recovery	n/a	ISS, AIS, Revised Trauma Score (RTS), GCS, hospital LOS, ICU LOS, presence of hollow viscus injusry involving urinary bladder or gastrointestinal tract, mortality	The American Surgeon
Richards et al., 2011, USA [43]	42.7 (SD = 16.8)	n/a	Number (%) Male (Study Group): 9 (50) Female (Study Group): 9 (50) Male (Control): 54 (60) Female (Control): 36 (40)	EC (orthopedic)—reamed intramedullary nailing	n/a	n/a	Global cognition, verbal and visual memory, visuospatial construction, processing speed, visual attention/verbal attention, executive functioning (verbal fluency and set shifting), and depressive and posttraumatic stress disorder symptoms	12 months after injury	Mini Mental State Exam, Rey Auditory Verbal Learning Test, Rey Osterreith Complex Figure Test–Delay, Rey Osterreith Complex Figure–Copy, Digit Symbol Coding, Trailmaking Test A, Digit Span, FAS, Trailmaking Test B Additionally, and Beck Depression Inventory-II and the Davidson Trauma Scale ISS, AIS	Journal of Orthopaedic Trauma
Prins et al., 2021, Muti-national [44]	Overall = 50 (37–63) SSRF = 50 (37–61) Nonoperative = 50 (37–63)	Overall = 26 (24–30) kg/m^2^ SSRF = 28 (25–31) kg/m^2^ Nonoperative = 26 (23–29) kg/m^2^	(male) Overall = 76.7% SSRF = 72.7% Nonoperative = 78%	Stabilization of rib fractures (SSRF)	n/a	n/a	Quality of life (postoperative recovery)	n/a	Ventilator free days, ICU-LOS, Hospital LOS, rate of/time to tracheostomy, occurrence of complications, GCS, mortality	Journal of Trauma Acute Care Surgery
Roberts et al., 2024, USA [45]	42.2 +/− 17.8 (SD)	n/a	Male: 1279 Female: 556	EC	n/a	n/a	Executive functioning, processing speed, memory, neurocognitive impairment	2 weeks and 6 months	GOSE-ALL, GOSE-TBI, Trail Making Test Parts A and B, WAIS-PSI, RAVLT Immediate Recall, and RAVLT-L, AIS, ISS	JAMA Surgery

## Data Availability

No new data were created.

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
