# Peer review of "Neurocognitive Outcomes After Extracranial Surgery and General Anesthesia in Patients with a History of Mild-to-Moderate Traumatic Brain Injury: Systemic Review and Meta-Analysis"

_biology, 2025, doi:10.3390/biology14060640_

Round 1
Reviewer 1 Report
Comments and Suggestions for Authors
Khan et al have provided exciting systematic review and meta-analysis study which summarizes the neurocognitive outcomes after extracranial surgery and GA in patients with a history of mild to Moderate TBI. Authors have presented strong methodology and provided comprehensive analysis of the results and published studies about matter. However, the manuscript can benefit from implementing of couple of minor comments:
- Could authors clarify please if there were language restrictions for the studies selections or only English studies were analyzed. If no, please add it to material and method section
- If possible, could authors please mention if prospective registration (PROSPERO) was or was not performed
- In the lines 183-186: Could authors please clarify the study selection process. Authors mention that from 27139 articles 8010 were removed, making it 19129 articles, from which automation tools marked 19129 studies as ineligible, which remains 0 articles. Please verify and clarify the selection steps. Also please verify how 3826 titles and abstracts were selected.
- Please standardize the citation format across the manuscript. For instance, in line 221 “Shapiro et al. 2001 [42]; Richards et al. 2011 [43]; Roberts et al., 2024” while at line 222 “(McKee et al., 1997) [41], and one (Prins et al. 2021) [44]” etc
- In line 248-249 authors mention that for TMT-A two studies assessed rote memory. Could authors please classify why those studies used rote memory, while TMT-A is considered to be a tests of speed for attention, sequencing, mental flexibility, visual search, and motor function.
- Line 255 please mention if MD 16.59 can be considered clinically relevant or not.
- Line 257. Please provide exact value of p-value (it can not be zero, for instance write p-value < 0.001)
- Lines 268-270, please avoid using the ‘statistically insignificant’, the standard terminology is ‘not statistically significant’ together with exact p-value can improve the mathematical rigor of the statement (see also lines 36 and 271)
- Lines 274-276, please provide the units for clarity (days)
- Lines 292-300 please provide effect size (MD+/- CI) to highlight the magnitude
- Lines 297-299 please replace p =0.00 with p<0.001
- Line 328 ‘Performance in TMA-A’ contains a typo, should be TMT-A
- In the discussion section please clarify the clinical importance of this study, further suggesting how it can be used in clinical practice.
Author Response
Reviewer #1:
Khan et al have provided exciting systematic review and meta-analysis study which summarizes the neurocognitive outcomes after extracranial surgery and GA in patients with a history of mild to Moderate TBI. Authors have presented strong methodology and provided comprehensive analysis of the results and published studies about matter. However, the manuscript can benefit from implementing of couple of minor comments:
Comment #1: Could authors clarify please if there were language restrictions for the studies selections or only English studies were analyzed. If no, please add it to material and method section
Response: There were no language restrictions. We have added a statement. Line 98: ‘The following databases were searched to retrieve articles from inception to April 2024 (Supplementary Table S1-5) without language restrictions: Ovid Medline, Ovid Embase, Ovid Emcare, Global Health, and APA PsycINFO.’
Comment #2: If possible, could authors please mention if prospective registration (PROSPERO) was or was not performed.
Response: Yes, the reference to the PROSPERO registration was included in the original version of the manuscript. Line 95: ‘The protocol of this systematic review and meta-analysis was registered at PROSPERO (CRD42024510980) and published prior to the review's completion [23].’
- Khan, Z. A., Sumsuzzman, D. M., Duran, T. A., Ju, L. S., Seubert, C. N., & Martynyuk, A. E. (2025). Perioperative Neurocognitive Disorder in Individuals with a History of Traumatic Brain Injury: Protocol for a Systematic Review and Meta-Analysis. Biology, 14(2), 197. https://doi.org/10.3390/biology14020197
Comment #3: In the lines 183-186: Could authors please clarify the study selection process. Authors mention that from 27139 articles 8010 were removed, making it 19129 articles, from which automation tools marked 19129 studies as ineligible, which remains 0 articles. Please verify and clarify the selection steps. Also please verify how 3826 titles and abstracts were selected.
Response: Thank you for identifying the error. We mistakenly recorded the number of ineligible studies as 19,303 instead of the correct figure, 15,303. The description of the method has been updated as follows (Line 187): ‘A total of 27,139 articles were retrieved from our selected databases using our curated search strategy (Supplementary Table S1). Utilizing the Rayyan screening tool, 8,010 studies were identified as duplicates and removed from the initial dataset. The remaining 19,129 studies were screened using the artificial intelligence tool ASReview [31,32]. Based on ASReview’s ranking of relevant articles, reviewers manually screened the top 20% of the most relevant studies by title and abstract, totaling 3,826 articles. Notably, more than 500 of the lowest-ranked articles within this set were found to be irrelevant according to the selection criteria (Figure 1). Of the 3,826 screened articles, 29 were selected for full-text review, while the remaining 15,303 (19,129 – 3,826 = 15,303) were deemed ineligible without title and abstract screening. The full text of these 29 selected articles was assessed against the selection criteria, resulting in the exclusion of 12 studies due to unrelated populations, 2 due to irrelevant study design, 6 due to irrelevant outcomes, and 4 due to the absence of an appropriate control group. Ultimately, five studies were included in the review [41-45]. Earlier studies have reported that screening only 15% of the most relevant articles is sufficient to successfully identify all relevant literature [31].’
We have also added details in the legend of figure 2.
‘Figure 2. The PRISMA 2020 flow diagram illustrates the process of identifying, screening, and including studies in a meta-analysis. A total of 27,139 articles were retrieved from our selected databases using our curated search strategy (Supplementary Table S1). Utilizing the Rayyan screening tool, 8,010 studies were identified as duplicates and removed from the initial set of 27,139 articles. The remaining 19,129 studies were screened using the artificial intelligence tool ASReview [31,32]. Based on ASReview’s ranking of relevant articles, reviewers manually screened the top 20% of the most relevant studies by title and abstract, totaling 3,826 articles. Of these, 29 were selected for full-text review, while the remaining 15,303 articles (19,129 – 3,826 = 15,303) were deemed ineligible without title and abstract screening. The full text of the 29 selected articles was assessed against the selection criteria, leading to the exclusion of 12 studies due to unrelated populations, 2 due to irrelevant study design, 6 due to irrelevant outcomes, and 4 due to the absence of an appropriate control group. Ultimately, five studies were included in the review [41-45].’
- Quan, Y.; Tytko, T.; Hui, B. Utilizing ASReview in screening primary studies for meta-research in SLA: A step-by-step tutorial. Research Methods in Applied Linguistics 2024, 3, 100101, doi:10.1016/j.rmal.2024.100101.
- Hindriks, S.J. A study on the user experience of the ASReview software tool for experienced and unexperienced users. 2020.
- McKee, M.D.; Schemitsch, E.H.; Vincent, L.O.; Sullivan, I.; Yoo, D. The effect of a femoral fracture on concomitant closed head injury in patients with multiple injuries. J. Trauma 1997, 42, 1041–1045, doi:10.1097/00005373-199706000-00009.
- Shapiro, M.B.; Nance, M.L.; Schiller, H.J.; Hoff, W.S.; Kauder, D.R.; Schwab, C.W. Nonoperative management of solid abdominal organ injuries from blunt trauma: impact of neurologic impairment. Am. Surg. 2001, 67, 793–796.
- Richards, J.E.; Guillamondegui, O.D.; Archer, K.R.; Jackson, J.C.; Ely, E.W.; Obremskey, W.T. The association of reamed intramedullary nailing and long-term cognitive impairment. J. Orthop. Trauma 2011, 25, 707–713, doi:10.1097/BOT.0b013e318225f358.
- Prins, J.T.H.; Van Lieshout, E.M.M.; Ali-Osman, F.; Bauman, Z.M.; Caragounis, E.-C.; Choi, J.; Christie, D.B.; Cole, P.A.; DeVoe, W.B.; Doben, A.R.; Eriksson, E.A.; Forrester, J.D.; Fraser, D.R.; Gontarz, B.; Hardman, C.; Hyatt, D.G.; Kaye, A.J.; Ko, H.-J.; Leasia, K.N.; Leon, S.; Marasco, S.F.; McNickle, A.G.; Nowack, T.; Ogunleye, T.D.; Priya, P.; Richman, A.P.; Schlanser, V.; Semon, G.R.; Su, Y.-H.; Verhofstad, M.H.J.; Whitis, J.; Pieracci, F.M.; Wijffels, M.M.E. Surgical stabilization versus nonoperative treatment for flail and non-flail rib fracture patterns in patients with traumatic brain injury. Eur. J. Trauma Emerg. Surg. 2022, 48, 3327–3338, doi:10.1007/s00068-022-01906-1.
- Roberts, C.J.; Barber, J.; Temkin, N.R.; Dong, A.; Robertson, C.S.; Valadka, A.B.; Yue, J.K.; Markowitz, A.J.; Manley, G.T.; Nelson, L.D.; Transforming Clinical Research and Knowledge in TBI (TRACK-TBI) Investigators; Badjatia, N.; Diaz-Arrastia, R.; Duhaime, A.-C.; Feeser, V.R.; Gopinath, S.; Grandhi, R.; Jha, R.; Keene, C.D.; Madden, C.; McCrea, M.; Merchant, R.; Ngwenya, L.B.; Rodgers, R.B.; Schnyer, D.; Taylor, S.R.; Zafonte, R. Clinical Outcomes After Traumatic Brain Injury and Exposure to Extracranial Surgery: A TRACK-TBI Study. JAMA Surg. 2024, 159, 248–259, doi:10.1001/jamasurg.2023.6374.
Comment #4: Please standardize the citation format across the manuscript. For instance, in line 221 “Shapiro et al. 2001 [42]; Richards et al. 2011 [43]; Roberts et al., 2024” while at line 222 “(McKee et al., 1997) [41], and one (Prins et al. 2021) [44]” etc
Response: Thank you for pointing out this error. We have updated our citation style as numbers [1,2,3…].’ These changes have been made throughout the manuscript.
Comment #5: In line 248-249 authors mention that for TMT-A two studies assessed rote memory. Could authors please classify why those studies used rote memory, while TMT-A is considered to be a tests of speed for attention, sequencing, mental flexibility, visual search, and motor function.
Response: Thank you for highlighting the important distinctions between rote memory and the cognitive abilities assessed by the Trail Making Test Part A (TMT-A). We agree with the reviewer that the TMT-A is not an ideal test for evaluating rote memory.
To address this issue, we have added the following text (Line 260): ‘Two studies reported using the TMT-A test to assess rote memory in individuals with TBI exposed to GA compared to unexposed individuals [41, 45]. In one study [41], the mean age of the subjects was 37.5 years at the time of evaluation, with a mean interval of 32 months between exposure and evaluation (minimum follow-up of 12 months). In the other study [45], the mean age was 42.2 years, with a 6-month interval between exposure and evaluation. In our meta-analysis of both studies, a significant difference was observed in the time required to complete TMT-A between exposed and unexposed TBI patients [I² = 0.0%, MD = 2.04; 95% CI (0.38, 3.70); p = 0.016] (Figure 4A). It is important to note that rote memory refers to the ability to recall information through repetition without necessarily understanding its meaning or context, whereas the primary purpose of the TMT-A is to evaluate processing speed, attention, and visual-motor coordination [47–49]. However, some investigators suggest that the processing speed, visual attention, sequencing, and motor function assessed by the TMT-A can be interpreted as features of rote memory, given that the test requires, for example, the repetitive and sequential connection of numbers under timed conditions [49,50].’
- McKee, M.D.; Schemitsch, E.H.; Vincent, L.O.; Sullivan, I.; Yoo, D. The effect of a femoral fracture on concomitant closed head injury in patients with multiple injuries. J. Trauma 1997, 42, 1041–1045, doi:10.1097/00005373-199706000-00009.
- Roberts, C.J.; Barber, J.; Temkin, N.R.; Dong, A.; Robertson, C.S.; Valadka, A.B.; Yue, J.K.; Markowitz, A.J.; Manley, G.T.; Nelson, L.D.; Transforming Clinical Research and Knowledge in TBI (TRACK-TBI) Investigators; Badjatia, N.; Diaz-Arrastia, R.; Duhaime, A.-C.; Feeser, V.R.; Gopinath, S.; Grandhi, R.; Jha, R.; Keene, C.D.; Madden, C.; McCrea, M.; Merchant, R.; Ngwenya, L.B.; Rodgers, R.B.; Schnyer, D.; Taylor, S.R.; Zafonte, R. Clinical Outcomes After Traumatic Brain Injury and Exposure to Extracranial Surgery: A TRACK-TBI Study. JAMA Surg. 2024, 159, 248–259, doi:10.1001/jamasurg.2023.6374.
- 4 Singer, M.; Fazaluddin, A.; Andrew, K.N. Recognition of categorised words: repetition effects in rote study. Memory 2013, 21, 467–481, doi:10.1080/09658211.2012.739625.
- Salthouse, T.A. What cognitive abilities are involved in trail-making performance? Intelligence 2011, 39, 222–232, doi:10.1016/j.intell.2011.03.001.
- Llinàs-Reglà, J.; Vilalta-Franch, J.; López-Pousa, S.; Calvó-Perxas, L.; Torrents Rodas, D.; Garre-Olmo, J. The trail making test. Assessment 2017, 24, 183–196, doi:10.1177/1073191115602552.
- O’Rourke, J.J.F.; Beglinger, L.J.; Smith, M.M.; Mills, J.; Moser, D.J.; Rowe, K.C.; Langbehn, D.R.; Duff, K.; Stout, J.C.; Harrington, D.L.; Carlozzi, N.; Paulsen, J.S. The Trail Making Test in prodromal Huntington disease: contributions of disease progression to test performance. J. Clin. Exp. Neuropsychol. 2011, 33, 567–579, doi:10.1080/13803395.2010.541228.
Comment #6: Line 255 please mention if MD 16.59 can be considered clinically relevant or not.
Response: We have added the following statement to the revised text (Line 279): ‘While there is no universally agreed-upon MD threshold for clinical significance, an MD of several seconds (e.g., 10–20 seconds or more) may indicate meaningful cognitive impairment in TBI populations [51].’
- Lange, R.T.; Iverson, G.L.; Zakrzewski, M.J.; Ethel-King, P.E.; Franzen, M.D. Interpreting the trail making test following traumatic brain injury: comparison of traditional time scores and derived indices. J. Clin. Exp. Neuropsychol. 2005, 27, 897–906, doi:10.1080/1380339049091290.
Comment #7: Line 257. Please provide exact value of p-value (it cannot be zero, for instance write p-value < 0.001)
Response: The p-value has been updated to p< 0.001.
Comment #8: Lines 268-270, please avoid using the ‘statistically insignificant’, the standard terminology is ‘not statistically significant’ together with exact p-value can improve the mathematical rigor of the statement (see also lines 36 and 271)
Response: The words ‘statistically insignificant’ are updated to ‘statistically not significant’.
Comment #9: Lines 274-276, please provide the units for clarity (days)
Response: We have added 'days' as the unit of measurement
Comment #10: Lines 292-300 please provide effect size (MD+/- CI) to highlight the magnitude
Response: We have added the MD and CIs.
Line 323: ‘however, heterogeneity remained high, with only slight reduction [MD = 3.43; CI [1.00, 5.85]; p = 0.01; I2 = 80.30%] compared to the original results [MD = 1.94; CI [-1.25, 5.13]; p = 0.137; I2 = 81.40%]. No single study exclusion reduced heterogeneity below 50%. Regarding hospital LOS, the p-value remained insignificant, but heterogeneity was eliminated [MD = 2.14; CIs [-0.38, 4.65], p<0.001; I2 = 0.00%] compared to the original results [MD= 0.99; CIs [-1.59, 3.57], p = 0.234; I2 = 38.60%].’
Comment #11: Lines 297-299 please replace p =0.00 with p<0.001
Response: Updated to p<0.001.
Comment #12: Performance in TMA-A’ contains a typo, should be TMT-A
Response: Thank you, this typographical error has been corrected.
Comment #13: In the discussion section please clarify the clinical importance of this study, further suggesting how it can be used in clinical practice.
Response: To further highlight the potential clinical and scientific significance of this systematic review and meta-analysis, which is based on a limited number of original studies, we have added the following statement to the revised text (Line 407):
‘We hope that this systematic review and meta-analysis will galvanize much-needed clinical and laboratory research on this clinically and scientifically significant topic, which has so far lacked sufficient attention, as evidenced by the identification of only five broadly relevant studies published in this field over the past 27 years.’
Reviewer 2 Report
Comments and Suggestions for Authors
Review Comments
The manuscript titled “Neurocognitive Outcomes after Extracranial Surgery and General Anesthesia in Patients with a History of Mild to Moderate Traumatic Brain Injury: Systematic Review and Meta-Analysis” presents a thorough and well-organized review of an important clinical issue- postoperative neurocognitive decline (PND) in patients with mild to moderate TBI. The authors conducted an extensive search across multiple databases and used standardized protocols to minimize bias. Overall, the work is timely and relevant. I have a few suggestions that could further strengthen the manuscript:
Major Comments:
- It would be helpful to include a table summarizing the patient demographics from the five studies used in the meta-analysis.
- Since both male and female patients were included, it would be interesting to discuss whether there was any gender-specific differences in neurocognitive outcomes.
- While the methods are clearly focused on reducing bias, the manuscript could benefit from placing a bit more emphasis on discussing the actual neurocognitive outcomes observed in these patients after surgery and anesthesia.
- A graphical abstract or schematic summarizing the study’s key findings would be beneficial.
- It would also be valuable to discuss the baseline neurocognitive status of patients after their initial TBI and how it compares to their postoperative outcomes. If there were differences among the five studies in this regard?
Minor Comments:
- Please review the manuscript for formatting issues, as several spacing errors were noted.
Author Response
Reviewer #2:
The manuscript titled “Neurocognitive Outcomes after Extracranial Surgery and General Anesthesia in Patients with a History of Mild to Moderate Traumatic Brain Injury: Systematic Review and Meta-Analysis” presents a thorough and well-organized review of an important clinical issue- postoperative neurocognitive decline (PND) in patients with mild to moderate TBI. The authors conducted an extensive search across multiple databases and used standardized protocols to minimize bias. Overall, the work is timely and relevant. I have a few suggestions that could further strengthen the manuscript:
Comment #1: It would be helpful to include a table summarizing the patient demographics from the five studies used in the meta-analysis.
Response: All available demographic data from the selected original studies were included in the original version of this manuscript (Table 1). The table presents the following information: gender, age (years), BMI/body weight (kg), type of surgery, anesthetics, route of administration, study participants' country, time elapsed between surgery and neurocognitive evaluation, evaluation method, neurocognitive outcomes, sample size, first author of the publication, year of publication, journal of publication, study funding, and authors' conflict of interest.
Comment #2: Since both male and female patients were included, it would be interesting to discuss whether there was any gender-specific differences in neurocognitive outcomes.
Response: We agree that the sex-dependent neurocognitive effects of TBI, surgery, and GA are critically important for both understanding the etiology of the disease and developing effective treatments. Although we had proposed a sex-based subgroup analysis in the previously published protocol for this study [1], the original studies that met the selection criteria for our meta-analysis do not provide sex-stratified data sets.
- Khan, Z.A.; Sumsuzzman, D.M.; Duran, T.A.; Ju, L.-S.; Seubert, C.N.; Martynyuk, A.E. Perioperative Neurocognitive Disorder in Individuals with a History of Traumatic Brain Injury: Protocol for a Systematic Review and Meta-Analysis. Biology (Basel) 2025, 14, doi:10.3390/biology14020197.
Comment #3: While the methods are clearly focused on reducing bias, the manuscript could benefit from placing a bit more emphasis on discussing the actual neurocognitive outcomes observed in these patients after surgery and anesthesia.
Response: We have added the following text in Line 368: ‘The neurocognitive outcomes observed in patients with TBI who underwent extracranial surgery under GA indicate significant impairments in various cognitive domains. Specifically, the delayed completion times in both TMT-A and TMT-B suggest deficits in cognitive flexibility, visual attention, and task-switching abilities [45]. Additionally, impairments in memory retention, language processing, and verbal fluency suggest that the effects of surgery and anesthesia may extend beyond executive functioning and cognitive flexibility.’
Furthermore, we have added the following text in Line 260:
‘Two studies reported using the TMT-A test to assess rote memory in individuals with TBI exposed to GA compared to unexposed individuals [41, 45]. In one study [41], the mean age of the subjects was 37.5 years at the time of evaluation, with a mean interval of 32 months between exposure and evaluation (minimum follow-up of 12 months). In the other study [45], the mean age was 42.2 years, with a 6-month interval between exposure and evaluation. In our meta-analysis of both studies, a significant difference was observed in the time required to complete TMT-A between exposed and unexposed TBI patients [I² = 0.0%, MD = 2.04; 95% CI (0.38, 3.70); p = 0.016] (Figure 4A). It is important to note that rote memory refers to the ability to recall information through repetition without necessarily understanding its meaning or context, whereas the primary purpose of the TMT-A is to evaluate processing speed, attention, and visual-motor coordination [47–49]. However, some investigators suggest that the processing speed, visual attention, sequencing, and motor function assessed by the TMT-A can be interpreted as features of rote memory, given that the test requires, for example, the repetitive and sequential connection of numbers under timed conditions [49,50].’
Comment #4: A graphical abstract or schematic summarizing the study’s key findings would be beneficial.
Response: We have added a graphical abstract and its legend.
Graphical abstract : We hypothesized that patients with a history of traumatic brain injury (TBI) may be more susceptible to general anesthesia (GA) and surgery-induced postoperative neurocognitive dysfunction (PND) due to the interaction of TBI- and GA/surgery-associated stress and inflammation. The findings of this systematic review and meta-analysis suggest that patients with mild to moderate TBI are at an increased risk of developing PND following extracranial surgery under GA. This review and meta-analysis also underscore the limited number of studies on PND in TBI patients, highlighting an urgent need for further clinical and preclinical research on this critical topic.
|
Comment #5: It would also be valuable to discuss the baseline neurocognitive status of patients after their initial TBI and how it compares to their postoperative outcomes. If there were differences among the five studies in this regard?
Response: We agree that assessing the baseline neurocognitive status of patients with TBI before their exposure to anesthesia or surgery would provide valuable insights into the neurocognitive effects of these procedures. However, the published sources we analyzed did not contain baseline neurocognitive data for TBI patients prior to anesthesia or surgery. We have incorporated this information into the revised manuscript.
Line 244: ‘None of the included studies contain baseline neurocognitive data for TBI patients prior to anesthesia or surgery.’
Comment #6: Please review the manuscript for formatting issues, as several spacing errors were noted.
Response: Thank you. We have corrected the formatting errors.

Reviewer 3 Report
Comments and Suggestions for Authors
Khan et al. present a systematic review and meta-analysis mentioning the possibility of neurocognitive decline following extracranial surgery under general anesthesia in patients with a history of mild to moderate traumatic brain injury.
This manuscript discusses an interesting compilation of studies that try to support the occurrence phenomenon described in the paper as postoperative neurocognitive disorder (PND).
My primary issue with this review and meta-analysis is the limited of studies that the authors were able to include in this paper, there were just 5 studies that fit the criteria for inclusion. In addition to that shortcoming, there was bias introduced in terms of demographics as 4 out of 5 of these studies were conducted on the North American continent, and the majority of participants in both experimental and control groups were male in 4 out of 5 studies. I realize that the authors have already acknowledged the lack of studies on postoperative neurocognitive disorders in traumatic brain injury patients and how there is an urgent need for further clinical and preclinical research, by inclusion of animal models. This unfortunately does not negate the fact that these shortcomings make the study weak. This is especially true when considering the actual data analysis and statistics that were performed. For example, for the trail making tests A and B, only two studies were included. The Glasgow Outcome Scale, similarly, was also performed on just two studies. These facts, although not surprising, considering different studies comprise of different assessments and comparisons, still make any conclusions harder to make. Although, I acknowledge the importance of studying this topic in order to realize the true disadvantages of using generalized anesthesia in these patients, the lack of studies make the impact of this manuscript limited and thus, it is unfit for publication.
In addition, I found the following technical issues:
- In section 3.2, it is not clear if the included studies contained within them data from patients with preexisting conditions, or were these patients included in the previous studies and excluded for this review.
- The last line of section 3.4.1 Has been carried over to the next page in the incorrect format.
- For figures, 4 and 5, please explain the P-values that are part of the figure as well as p-values written over the figure and the difference between them.
- The last paragraph. Has been repeated twice, from lines 381 to 387 and again from lines 388 to 394, with different references.
Author Response
Reviewer #3: Khan et al. present a systematic review and meta-analysis mentioning the possibility of neurocognitive decline following extracranial surgery under general anesthesia in patients with a history of mild to moderate traumatic brain injury.
This manuscript discusses an interesting compilation of studies that try to support the occurrence phenomenon described in the paper as postoperative neurocognitive disorder (PND).
Comment #1: My primary issue with this review and meta-analysis is the limited of studies that the authors were able to include in this paper, there were just 5 studies that fit the criteria for inclusion. In addition to that shortcoming, there was bias introduced in terms of demographics as 4 out of 5 of these studies were conducted on the North American continent, and the majority of participants in both experimental and control groups were male in 4 out of 5 studies. I realize that the authors have already acknowledged the lack of studies on postoperative neurocognitive disorders in traumatic brain injury patients and how there is an urgent need for further clinical and preclinical research, by inclusion of animal models. This unfortunately does not negate the fact that these shortcomings make the study weak. This is especially true when considering the actual data analysis and statistics that were performed. For example, for the trail making tests A and B, only two studies were included. The Glasgow Outcome Scale, similarly, was also performed on just two studies. These facts, although not surprising, considering different studies comprise of different assessments and comparisons, still make any conclusions harder to make. Although, I acknowledge the importance of studying this topic in order to realize the true disadvantages of using generalized anesthesia in these patients, the lack of studies make the impact of this manuscript limited and thus, it is unfit for publication.
Response: We agree with the reviewer that our systematic review and meta-analysis would be stronger if more original studies meeting the selection criteria were conducted in this field. We appreciate the reviewer’s acknowledgment of one of the main conclusions of this review, which highlights “an urgent need for further clinical and preclinical research” in the field of neurocognitive effects of anesthesia/surgery in patients with a history of TBI. However, we respectfully disagree with several of the reviewer’s specific comments. For example, the reviewer states, “My primary issue with this review and meta-analysis is the limited of studies that the authors were able to include in this paper...” The issue is not “the limited of studies that the authors were able to include in this paper,” but rather the limited number of studies that have been conducted in this field.
Furthermore, the following statement by the reviewer may imply that the authors of this systematic review and meta-analysis introduced bias related to demographic data: “In addition to that shortcoming, there was bias introduced in terms of demographics as 4 out of 5 of these studies were conducted on the North American continent, and the majority of participants in both experimental and control groups were male in 4 out of 5 studies.” However, the authors analyzed all identified studies rather than selectively choosing among them, and therefore, did not introduce any bias. The bias in this field arises from the limited number of studies on this topic, which further supports the need for more comprehensive research.
We also respectfully disagree with the reviewer’s statement: “I acknowledge the importance of studying this topic in order to realize the true disadvantages of using generalized anesthesia in these patients.” The true importance of studying this topic lies in its potential to help develop safer anesthesia/surgery regimens and/or pre-anesthesia/surgery pharmacotherapeutic approaches to alleviate or prevent neurocognitive deficits after anesthesia/surgery in patients with a history of TBI.
The value of this systematic review and meta-analysis lies in its findings on the interaction between neurocognitive effects of TBI and anesthesia/surgery, as well as its emphasis on the scarcity of research in this field and the urgent need for further investigation. We hope that this systematic review and meta-analysis will draw attention to this clinically and scientifically significant topic, which has generated only five broadly relevant studies over the past 27 years. By deeming this study “unfit for publication,” the reviewer risks allowing this field to remain dormant rather than encouraging much-needed progress.
Comment #2: In section 3.2, it is not clear if the included studies contained within them data from patients with preexisting conditions, or were these patients included in the previous studies and excluded for this review.
Response: The included studies excluded the data from patients with preexisting conditions. We have updated the statement in section 3.2 for clarity.
Line 245: ‘Regarding confounding variables or preexisting conditions, the included studies excluded patients with prior neurocognitive deficiencies or a history of TBI.’
Comment #3: The last line of section 3.4.1 Has been carried over to the next page in the incorrect format.
Response: Thank you. We have corrected the formatting errors.
Comment #4: For figures, 4 and 5, please explain the P-values that are part of the figure as well as p-values written over the figure and the difference between them.
Response: We have added an explanation of two p-values in the legend section of both figures: ‘The p-values next to I2 on the right hand of the forest plot represents significance of heterogeneity, while p-value on the left hand shows significance between treatment and control groups.’
Comment #5: The last paragraph. Has been repeated twice, from lines 381 to 387 and again from lines 388 to 394, with different references.
Response: Thank you. We have corrected these errors.
Round 2
Reviewer 3 Report
Comments and Suggestions for Authors
I acknowledge all the corrections to text made in the manuscript based on my comments. They help increase the understanding of material and data presented in the manuscript.
Regarding the impact of this meta-analysis and systematic review, I still stand-by my earlier comment regarding the value this adds to encouraging future research in the field being limited by the fact that there weren’t enough studies that could be included in this manuscript. Meta-analyses are most compelling when they provide comprehensive, quantitative insights that can shift current understanding or practice, this manuscript, currently, is unable to do so due to a lack of a compilation of broadly relevant studies.